# Quercetin Inhibits Colorectal Cancer Cells Induced-Angiogenesis in Both Colorectal Cancer Cell and Endothelial Cell through Downregulation of VEGF-A/VEGFR2

Tamonwan Uttarawichien [1], Chantra Kamnerdnond [1], Tasanee Inwisai [1], Prasit Suwannalert [1], Nathawut Sibmooh [2] and Witchuda Payuhakrit [1,*]

[1] Department of Pathobiology, Faculty of Science, Mahidol University, Bangkok 10400, Thailand; book.tamonwan@hotmail.com (T.U.); k.chantra.12@gmail.com (C.K.); tasanee.inw@mahidol.ac.th (T.I.); prasit.suw@mahidol.ac.th (P.S.)

[2] Faculty of Medicine Ramathibodi Hospital, Chakri Naruebodindra Medical Institute, Mahidol University, Samut Prakan 10540, Thailand; nathawut.sib@mahidol.ac.th

* Correspondence: witchuda.pay@mahidol.ac.th

**Abstract:** Colorectal cancer (CRC) aggressiveness is caused by cancer angiogenesis which promotes the cancer growth and metastasis associated with poor prognosis and poor survival. The vascular endothelial growth factor-A (VEGF-A) and its receptor (VEGFR-2) form the major signaling pathway in cancer angiogenesis. This study aimed to investigate the anti-angiogenesis activity of quercetin in both colorectal cancer cells and endothelial cells. The tube formation of human vein endothelial cells (HUVECs) was determined by using conditioned media of HT-29 cells treated with quercetin co-cultured with HUVECs. The VEGF-A and NF-κB p65 protein expressions in the quercetin-treated HT-29 cells were determined by fluorescence assay and Western blot analysis. The VEGFR-2 protein expression in HUVECs was determined after they were co-cultured with the quercetin-treated HT-29 cells. Quercetin markedly decreased the HT-29 cell-induced angiogenesis in HUVECs. NF-κB p65 and VEGF-A protein expression were also inhibited by quercetin. Moreover, quercetin significantly inhibited VEGFR-2 expression and translocation in HUVECs after they were co-cultured with high dose quercetin-treated HT-29 cells. Taken together, quercetin had an anti-angiogenesis effect on VEGF-A inhibition related to the NF-κB signaling pathway in the HT-29 cells and reduced VEGFR-2 expression and translocation in HUVECs.

**Keywords:** colorectal cancer; angiogenesis; quercetin; NF-κB; VEGF-A; VEGFR-2

## 1. Introduction

Colorectal cancer (CRC) is currently the third most common cause of cancer mortality worldwide with more than 1.85 million cases and 850,000 deaths annually [1]. The occurrence of CRC is strongly associated with lifestyle habits, including unhealthy diet, physical inactivity, and the high consumption of alcohol and tobacco [1,2]. Other risk factors related to the onset and progression of CRC are complications of inflammatory bowel disease, a chronic intestinal inflammation, which is known to encourage carcinogenesis in combination with genetic factors in the sequence of inflammation–dysplasia–carcinoma, such as ulcerative colitis and Crohn's disease [3,4].

CRC aggressiveness is caused by cancer angiogenesis, a type of neovascular formation which is essential for the normal physiological functions of tissues, such as wound healing and embryonic development, but which is also an important factor in cancer progression due to cancer cells requiring it to provide them with proper nourishment and the removal of metabolic wastes, without which, cancer cells would die [5]. However, angiogenesis not only promotes cancer proliferation but also facilitates cancer metastasis and increases chemotherapeutic drug resistance [6,7]. There are two regulations of angiogenic factors involved in cancer angiogenesis—up-regulation of angiogenic activators and

down-regulation of their inhibitors—which is called an angiogenic switch [8,9]. Vascular endothelial growth factor (VEGF) is one of the important pro-angiogenic signals and is composed of five different glycoproteins of the VEGF family: VEGF-A, VEGF-B, VEGF-C, VEGF-D, and placental growth factor [10]. Previous studies have reported that the binding of VEGF-A to VEGFR-2 is mainly potent in the angiogenic mechanism [11]. The expression of VEGF is increased by being induced both dependently and independently from hypoxia-inducible transcription factor (HIF) mechanisms through the activation of the NF-κB pathway in endothelial cells; when NF-κB is activated, it will translocate into the nucleus and up-regulate the angiogenic signals which are necessary for the vascularization process [12–16]. One recent study by Lei Yin et al. also showed the expression and significance of VEGF in CRC patients, which were analyzed by using immunohistochemistry analysis in the patients. Their study showed that HIF-1α and VEGF were overexpressed in both the primary and matched metastatic tissues of the CRC. Therefore, inhibiting VEGF-A is needed for inhibiting both primary and metastatic CRC [17].

Quercetin (3,3′,4′,5,7-pentahydroxyflavone) is a dietary flavonoid compound found in various fruits and vegetables, such as onion, buckwheat, and broccoli [18]. It has several potential benefits for human health, such as antioxidant, anti-inflammatory, and anti-cancer agents, which make it act as an anti-cancer agent through broad-spectrum mechanisms from anti-carcinogenesis to anti-metastasis [19–21]. Several studies have reported that quercetin has an anti-angiogenic effect in prostate cancer, breast cancer, retinoblastoma, and head and neck carcinoma [22–27]. However, the precise anti-angiogenic effect of quercetin remains unclear. Therefore, the present study aimed to investigate the angiogenic effect of quercetin in HT-29 cell-induced angiogenesis in both colorectal and endothelial cells through the VEGF-A/VEGFR-2 pathway.

## 2. Materials and Methods

### 2.1. Chemicals and Reagents

Quercetin was purchased from Sigma (St. Louis, MO, USA), dissolved in dimethyl sulfoxide (DMSO), and stored at −20 °C. Matrigel was purchased from Corning, USA. The antibodies—anti-VEGF-A and anti-β–actin—were purchased from Merck, Germany, while anti-VEGFR-2 was purchased from GeneTex, North America (Alton Pkwy Irvine, CA, USA). Anti-NF-κB p65 antibody was purchased from Abcam (Cambridge, UK).

### 2.2. Cell Culture

HT-29 cells and human umbilical vein endothelium cells (HUVECs) was purchased from American Type Culture Collection (ATCC). HT-29 cells were cultured in completed Dulbecco's Modified Eagle Medium-F12 supplemented with 10% heat-inactivated fetal bovine serum, 1% L-glutamine, and 1% penicillin-streptomycin. Human umbilical vein endothelium cells (HUVECs) were maintained in complete endothelial cell growth medium supplemented with 2% fetal calf serum and growth factor supplements from the endothelial cell growth medium 2 kit (PromoCell, Heidelberg, Germany). The cells were maintained at 37 °C in a humidified atmosphere within an incubator that was supplied with 5% $CO_2$.

### 2.3. Conditioned Media and Co-Cultured Experiment

The HT-29 cells were cultured in 6-well plates until 80% confluence was reached. Then, the media were changed to serum-free media, and the conditioned media were harvested after 24 h of incubation. After the conditioned media were centrifuged, they were filtered through a 0.02 μm filter and then stored at −20 °C for use in tubulogenesis assay. For the co-culture in the proteins expression study, HUVECs $5 \times 10^4$ cells were seeded on coverslips and placed at the bottom of 6-well plates for IFA or seeded directly on 6-well plates for western blot analysis. They were incubated at 37 °C and 5% $CO_2$ for 48 h. Then the HUVEC cells were co-cultured with the HT-29 cells in polycarbonate Membrane Transwell® Inserts with 3 μm pore for 24 h.

### 2.4. MTT Assay

The cell proliferation of the HT-29 and HUVEC cells were evaluated by using a 3-(4,5-dimethythiazol-2-yl)-2,5 diphenyltetrasodium bromide (MTT) reagent. For the HT-29 cells, $1.5 \times 10^4$ cells, and for the HUVEC, $1.0 \times 10^4$ cells were seeded in a 96-well plates and incubated at 37 °C in a humidified atmosphere with 5% $CO_2$ for 24 h. Then, the media were changed to serum-free media, which contained quercetin at various concentrations and incubated for 24 h. The old media were replaced by 100 μL of media, which contained MTT solution, and incubated for 2 h before 100 μL of DMSO was added. The results were detected at 570 nm with a microplate reader (1420 victor2, Wallac (Boston, MA, USA)).

### 2.5. Tubulogenesis Assay

Matrigel solution was added into 96-well plates and incubated at 37 °C for 30 min. For the HUVEC cells, $8 \times 10^3$ cells were resuspended in the HT-29 conditioned media with or without 5 and 10 μg/mL of quercetin. Then, the HUVEC cells, $8 \times 10^3$ cells, were seeded onto a layer of Matrigel and incubated for 6 h. Tubular structures on the Matrigel were photographed from 3 randomly chosen fields. The total length of each tube per area was measured and analyzed by Image J software with an angiogenic analyzer.

### 2.6. Indirect Immunofluorescence Assay

Indirect immunofluorescence (IFA) was used to measure NF-κB p65 and VEGF-A expression in the HT-29 cells and VEGFR-2 expression in HUVECs. For the HT-29 cells, $4 \times 10^4$ cells were seeded on coverslips and placed at the bottom of 6-well plates. They were incubated at 37 °C with 5% $CO_2$ for 48 hours, after which, serum-free media containing 5 or 10 μg/mL quercetin were added and then incubated for another 24 h. The HT-29 cells were fixed with cold methanol, permeabilized with 0.25% Triton X-100, and then a primary antibody; including anti-NF-kB (1:1000), anti-VEGF-A (1:1000), and anti-VEGFR-2 (1:1000) was added. This was then incubated for 1.5 hours before a secondary antibody was added and incubated for another 30 min. Hoechst-33342 in dilution 1:500 was used for counterstaining for 15 min. For the HUVECs, $5 \times 10^4$ cells were seeded on coverslips and co-cultured with HT-29 cells as previously described. Then, the coverslips of HUVEC cells were harvested and fixed for immunostaining as previously described as above. The cells were observed under a fluorescence microscope (Olympus BX53, Japan) at the excitation and emission wavelength of 490/515 nm and the results are presented as the mean intensity of fluorescence that was analyzed by Image J program of 3 random fields in triplicate.

### 2.7. Western Blot Analysis

Total protein was obtained from the HT-29 cells treated with quercetin at concentrations of 5 and 10 μg/mL by using a cold RIPA buffer and scratched the cells. Then, the protein extracts were collected and centrifuged with 4 °C and 12,000 rpm. The supernatants were collected and measured protein concentration by using the Bradford assay. Then, NF-κB p65 and VEGF-A were detected by the Jess Simple Western System, a ProteinSimple automated Western blot system, under the principle of Western blot analysis with a specific capillary vacuum system in accordance with the instructions. Briefly, lysate proteins 2 μg were loaded for separating and then transferring in the capillaries containing the matrix gel. Afterwards, the surface was blocked and then probed with primary antibodies; including anti-NF-kB (1:1000) and anti-VEGF-A (1:1000) and then detected with HRP-conjugated secondary antibodies. The signals were developed, and the image was acquired for the pattern of protein separation according to molecular weight. β-actin was used as a loading control.

### 2.8. Statistical Analysis

The values were presented as mean ± SD from three independent experiments. One-way analysis of variance (ANOVA) was used for statistical analysis by using GraphPad Prism version 9. $p < 0.05$ was considered significantly different.

## 3. Results

### 3.1. Effect of Quercetin on Cell Proliferation in HT-29 and HUVEC Cells

The results demonstrated that quercetin at concentrations of 0.1, 1, and 10 µg/mL decreased HT-29 cell proliferation to 98.91 ± 3.84%, 92.50 ± 3.53%, and 93.93 ± 2.45%, respectively, while quercetin at a concentration of 100 µg/mL significantly decreased HT-29 cell proliferation to 70.90 ± 1.88% when compared with untreated cells ($p < 0.001$) as shown in Figure 1a. Similarly, quercetin at concentrations of 0.1, 1, and 10 µg/mL decreased HUVEC cell proliferation to 98.12 ± 0.31%, 93.73 ± 4.70%, and 85.58 ± 2.19%, respectively, while quercetin at a concentration of 100 µg/mL significantly decreased HUVEC cell proliferation to 65.83 ± 1.25% when compared with untreated cells ($p < 0.001$) as shown in Figure 1b. These results demonstrate that, at a high dose, quercetin inhibits cells proliferation, whereas low concentrations have no anti-proliferative effect on either cells type.

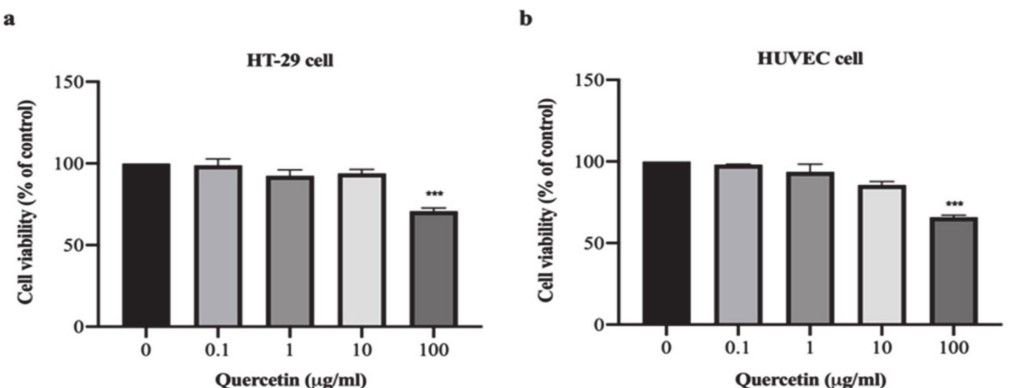

**Figure 1.** Effect of quercetin on cell proliferation in HT-29 and HUVEC cells. Cell proliferation of HT-29 and HUVEC cells treated with quercetin at various concentrations for 24 h was determine using MTT assay. The cell viability of HT-29 cells (**a**) and HUVEC cells (**b**) Bar graph were represented as mean ± SD, *n* = 3, *** $p < 0.001$.

### 3.2. Quercetin Significantly Inhibits HT-29 Cells-Induced Tube Formation of HUVECs

The tube formation on the HT-29 cell-induced HUVECs greatly increased with more tubular branching and elongated shapes. In contrast, the HUVECs treated with quercetin at concentrations of 5 and 10 µg/mL showed little tubular branching or elongated shapes as shown in Figure 2a. After analyzing the total tube length, quercetin at concentrations of 5 and 10 µg/mL was found to significantly decrease tube formation to 27.60 ± 2.88 and 23.42 ± 3.68, respectively when compared with the control (84.75 ± 1.68; $p < 0.001$) as shown in Figure 2b. These results indicate that HT-29 cells have the capability to induce tube formation on HUVECs while quercetin dramatically inhibits HT-29 cell-induced tube formation in a dose-dependent manner.

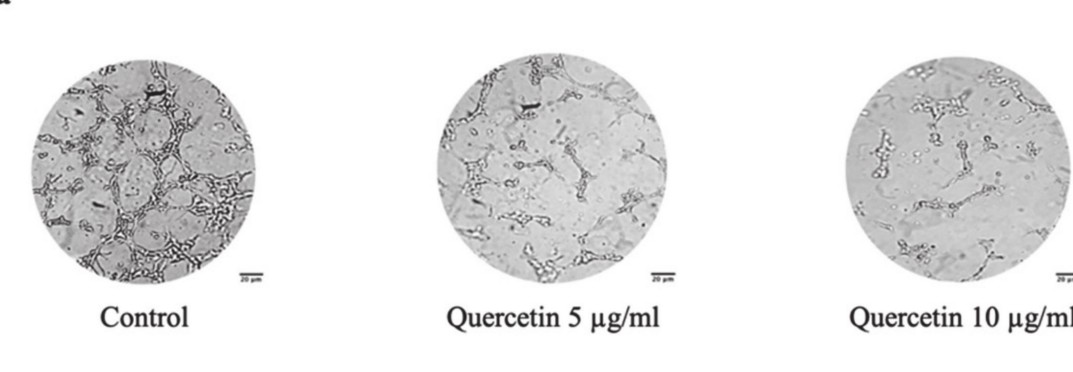

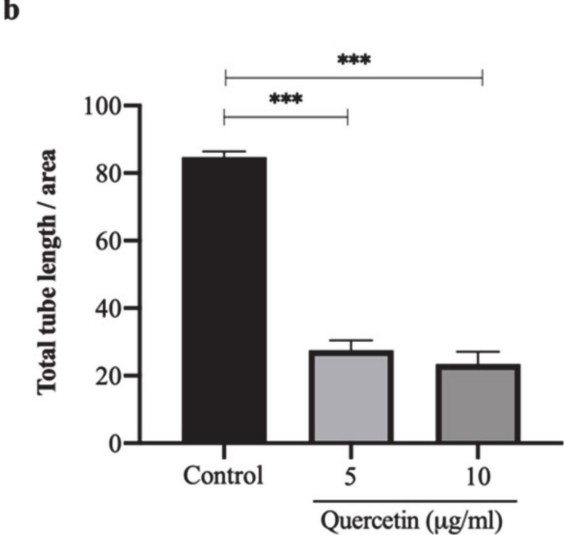

**Figure 2.** Quercetin inhibits HT-29 cells-induced tube formation of HUVECs. HUVEC cells were incubated with HT-29 cells conditioned media with or without quercetin for 6 h. The pictures revealed tubular morphology of HUVECs on the Matrigel layer (**a**) Scale bar: 20 μm. Total tube length was measured and analyzed by ImageJ program with angiogenic analyzer which represented as bar graph (**b**), mean ± SD, *n* = 3, *** $p < 0.001$.

### 3.3. Quercetin Significantly Inhibits NF-κB p65 Expression in HT-29 Cells

NF-κB p65 protein was mainly localized at the perinuclear and intranuclear regions. After treatment with quercetin at concentrations of 5 and 10 μg/mL, NF-κB p65 protein was observed in the cytoplasm, as shown in Figure 3a. The percentage of fluorescent intensity of NF-κB p65 expression in the quercetin-treated cells was markedly decreased to 60.77 ± 1.44% and 53.96 ± 8.80% at concentrations of 5 and 10 μg/mL, respectively, when compared with the control (100.00 ± 2.30%, $p < 0.001$) as shown in Figure 3b. Correlating with and confirming the Western blot analysis, quercetin significantly decreased NF-κB p65 protein expression in a dose-dependent manner as shown in Figure 3c,d. These findings indicate that quercetin inhibited the expression of NF-κB p65 protein in HT-29 cells.

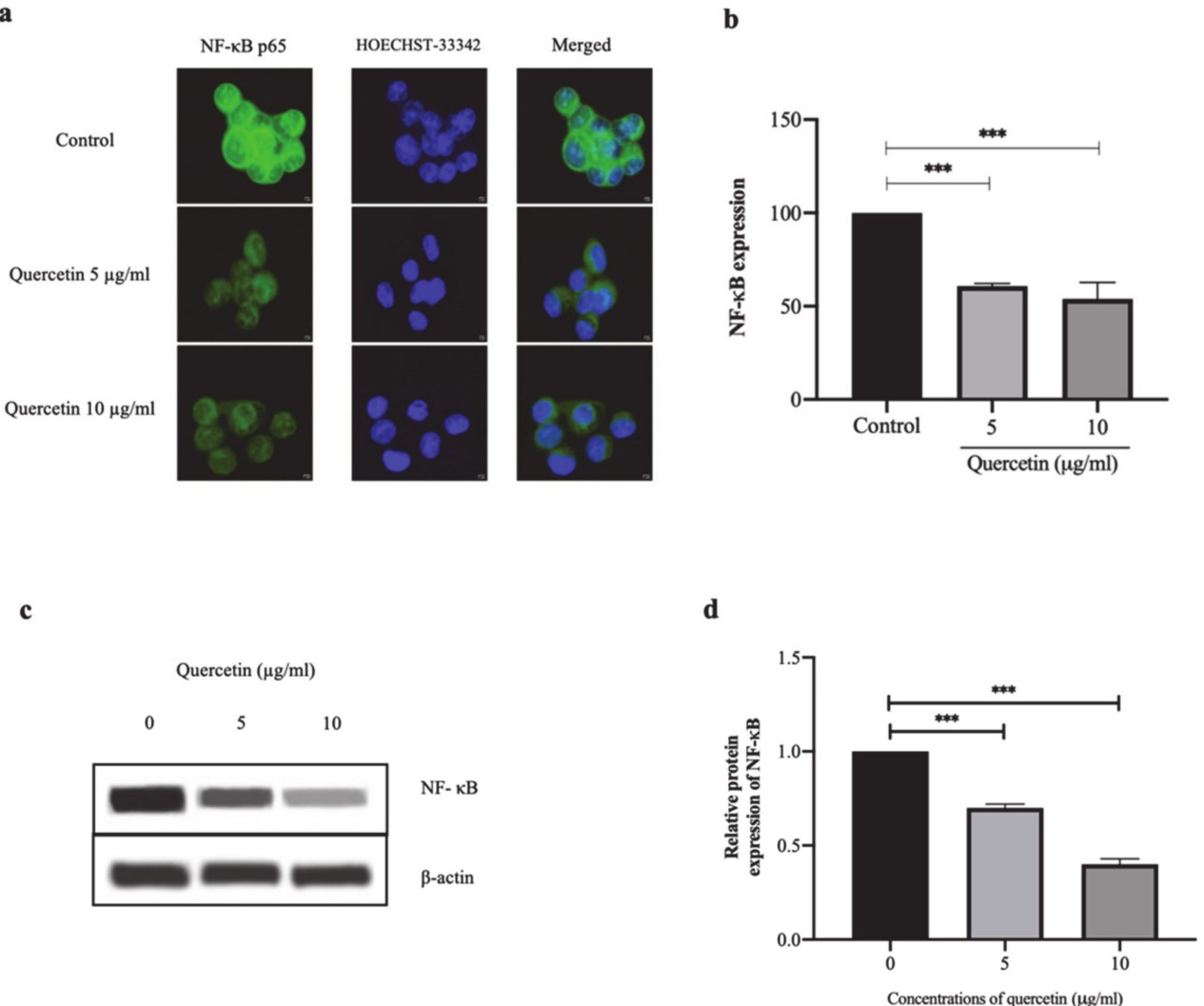

**Figure 3.** Quercetin significantly inhibits NF-κB p65 expression in HT-29 cells. In HT-29 cells treated with quercetin at concentrations 5 and 10 µg/mL for 24 h, the localization and expression of NF-κB p65 by immunofluorescence assay was investigated. (**a**) Immunofluorescence images of NF-κB p65 as represented in green and Hoechst-33342 as represented in blue for nuclear staining. Scale bar: 5 µm. (**b**) The fluorescent intensity of NF-κB p65 expression was calculated and presented as % of control. (**c**) Western blot analysis of NF-κB p65 expression. (**d**) Relative protein expression of NF-κB (normalized to ß-actin). Bar graph expressed as means ± SD, *n* = 3, *** *p* < 0.001.

### 3.4. Quercetin Significantly Suppresses VEGF-A Protein Expression in HT-29 Cells

HT-29 cells were treated with quercetin at concentrations of 5 and 10 µg/mL for 24 h. Then, IFA and Western blot analysis were used to investigate the VEGF-A protein expression. The IFA staining of VEGF-A protein expression was observed in the cytoplasm of HT-29 cells as shown in Figure 4a. When the HT-29 cells were treated with 5 µg/mL of quercetin, VEGF-A protein expression was significantly reduced to 80.48 ± 6.48% ($p < 0.01$), whereas when the HT-29 cells were treated with 10 µg/mL of quercetin, the expression of VEGF-A protein was significantly suppressed when compared with the control (100 ± 7.54%, $p < 0.001$) as shown in Figure 4b. Western blot analysis also showed that quercetin significantly reduced the expression of VEGF-A protein, as shown in Figure 4c,d.

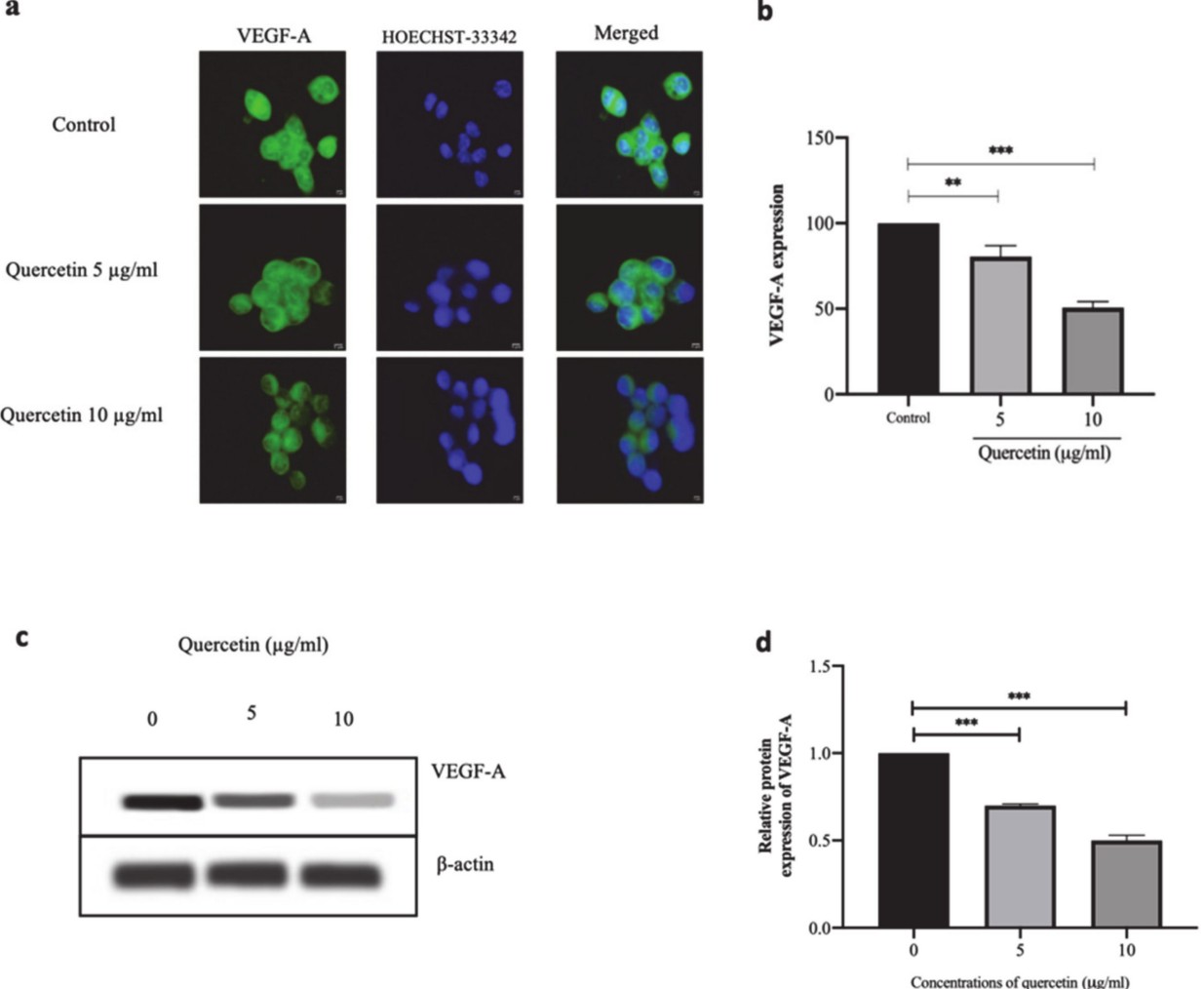

**Figure 4.** Quercetin significantly suppresses VEGF-A protein expression in HT-29 cells. In HT-29 cells treated with quercetin at concentrations 5 and 10 μg/mL for 24 hours, the localization and expression of VEGF-A protein by immunofluorescence assay was investigated. (**a**) Immunofluorescence images of VEGF-A as represented in green and Hoechst-33342 as represented in blue for nuclear staining. Scale bar: 5 μm. (**b**) The fluorescent intensity of VEGF-A expression was calculated and presented as % of control. (**c**) Western blot analysis of VEGF-A expression. (**d**) Relative protein expression of VEGF-A (normalized to ß-actin). Bar graph expressed as means ± SD, *n* = 3, ** *p* < 0.01, *** *p* < 0.001.

### 3.5. Quercetin Significantly Decreased VEGFR-2 Expression in HT-29 Cells-Induced HUVECs

VEGFR-2 expression was determined in HUVEC cells co-cultured with HT-29 cells treated with or without quercetin by using IFA. The HUVEC cells co-cultured with HT-29 cells showed a presence of green spots of VEGFR-2 in the cytoplasm, while the co-cultured cells treated with quercetin VEGFR-2 were mainly located in the nucleus (Figure 5a). The fluorescent intensity was analyzed, which showed that the fluorescence intensity of VEGFR-2 was decreased to 96.01 ± 4.42% and 79.11 ± 3.89% in HUVEC cells co-cultured with quercetin at concentrations of 5 and 10 μg/mL, respectively, compared with 100 ± 4.49% in the control (Figure 5b). These findings suggest that, at a concentration of 10 μg/mL, quercetin significantly decreased VEGFR-2 expression and translocation in HT-29 cell-induced HUVECs.

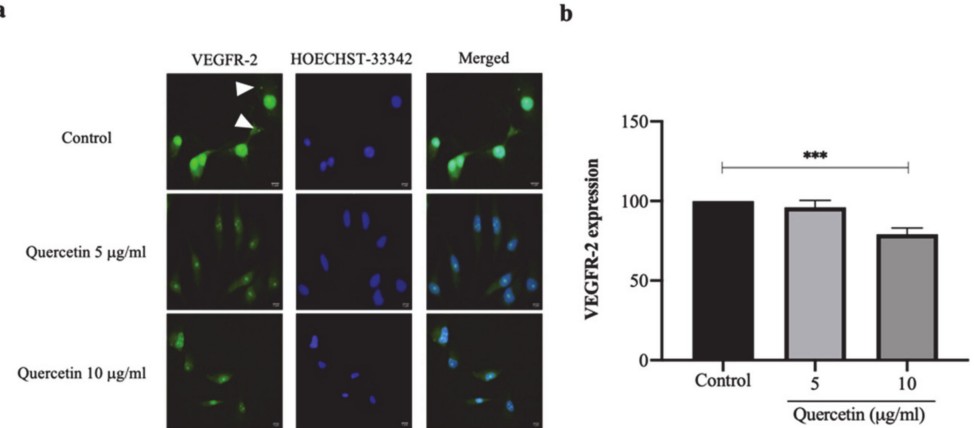

**Figure 5.** Quercetin significantly decreased VEGFR-2 expression in HT-29 cells-induced HUVECs. HUVEC cells were co-cultured with HT-29 cells with or without quercetin in polycarbonate Membrane Transwell® Inserts with 3 µm pore for 24 h and observed VEGFR-2 expression by immunofluorescence assay. (**a**) Immunofluorescence images of VEGFR-2 as represented in green and localization as shown in white arrowhead. Hoechst-33342 as represented in blue for nuclear staining. Scale bar: 5 µm. (**b**) The fluorescent intensity of VEGFR-2 expression was calculated and presented as % of control. Bar graph expressed as means ± SD, $n = 3$, *** $p < 0.001$.

## 4. Discussion

Our findings showed the tube formation in HT-29 cell-induced HUVECs was markedly inhibited by quercetin in a dose-dependent manner, which indicates the anti-angiogenic activity of quercetin. The present study concurred with previous studies that quercetin is an anti-angiogenesis agent in various types of cancer, including colorectal cancer Our results showed quercetin 10 ug/mL effectively suppressed angiogenesis as similar concentration with previously observed in breast cancer, retinoblastoma, and nasopharyngeal carcinoma [23–25,27]. Zhao et al. also demonstrated the anti-angiogenic activity of quercetin in transgenic zebrafish embryos and in HUVECs, with their findings showing that quercetin inhibited cell viability, VEGFR-2 expression, and tube formation in a dose-dependent manner [26]. In CRC, activated NF-κB is found in increasing amounts, with previous studies revealing the greater roles of NF-κB as a main regulator of VEGF and IL-8 expression in cancer angiogenesis [28–32]. Our findings also showed the effect of quercetin on suppressing both NF-κB p65 and VEGF-A expression in HT-29 cells. These results suggest that the anti-angiogenic effect of quercetin is caused by inhibited VEGF-A expression, which is the main mediator of the angiogenic process. Suppressing VEGF-A expression might be caused by the inhibitory effects of the NF-κB signaling pathway because VEGF-A is known to be one of the target genes of NF-κB activation [33].

In addition, quercetin inhibited angiogenesis, not only in colorectal cancer cells but also in endothelial cells. Our findings showed that quercetin inhibits VEGFR-2, the specific receptor for VEGF-A located on endothelial cells, which is the major molecule associated with initiating the cancer angiogenesis process through proliferating, migrating, and differentiating regulation of endothelial cells [34,35]. Our findings showed that high dose quercetin significantly inhibited VEGFR-2 expression in HUVEC, which is in line with the findings of a previous study that reported the angiogenesis inhibition of quercetin through targeting VEGFR-2 expression [27]. Moreover, our study also showed HT-29 cell-induced angiogenesis via the upregulation of VEGFR-2 on the cell surface of HUVECs, which corelates with a previous study by Liu et al., who observed the role of VEGFR-2 in promoting the morphologically and functionally endothelial differentiation of CRC [34].

Interestingly, our findings showed that quercetin also inhibited the translocation of VEGFR-2 into the surface of HUVECs. This concurs with the findings of Abhinand CS et al., who noted that the angiogenesis and the function of endothelial cells are regulated by recycling VEGFR-2 from endosome to plasma membrane. The binding between VEGF-

A and the surface of endothelial cells stimulates VEGFR-2 trafficking from the Golgi apparatus to the plasma membrane [35]. Moreover, from previous research, the increased expression of VEGFR-2 correlated with differentiation, metastasis/recurrence, and poor prognosis in human colon cancer samples [17]. These findings suggest that VEGFR-2 is functional at the surface of the endothelial cell and VEGFR-2 has the potential as a molecule to be used for anti-angiogenesis cancer therapy. Taken together, the anti-angiogenic activity of quercetin could be an angiogenesis inhibitor that directly inhibits the translocation and expression of VEGFR-2 in endothelial cells in colorectal cancer cell-induced angiogenesis. The particular localization of colorectal cancer in the digestive tract enables the achievement of significantly higher doses of quercetin at the cancer site, which is an advantage considering the low bioavailability of this compound.

**5. Conclusions**

In conclusion, our findings demonstrated that quercetin inhibited HT-29 colorectal cancer cell-induced angiogenesis through the suppression of VEGF-A related with NF-κB p65 expression in HT-29 cells. Moreover, quercetin inhibited both the translocation and the expression of VEGFR-2 in HUVECs, although clarification is still needed to determine the certain mechanisms of the inhibitory effect of quercetin, which could then be used in targeted therapy for the treatment of colorectal cancer with less adverse effects.

**Author Contributions:** Investigation and writing draft manuscript, C.K. and T.U.; visualization, T.U.; investigation and resources, T.I.; supervision, P.S. and N.S.; conceptualization, writing review and editing, W.P. All authors have read and agreed to the published version of the manuscript.

**Funding:** This research is funded by Faculty of Science, Mahidol University.

**Institutional Review Board Statement:** Not applicable.

**Informed Consent Statement:** Not applicable.

**Data Availability Statement:** All data presented or analyzed during this study are included in the article.

**Acknowledgments:** This study would not be accomplished without grants from the Faculty of Science, Mahidol University. In western blot analysis, JESS model of ProteinSimple automated western blot system and the test kit were kindly supported by N.Y.R. Limited Partnership, Thailand.

**Conflicts of Interest:** The authors declare no conflict of interest.

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
