# Peer review of "Quercetin Inhibits Colorectal Cancer Cells Induced-Angiogenesis in Both Colorectal Cancer Cell and Endothelial Cell through Downregulation of VEGF-A/VEGFR2"

_scipharm, doi:10.3390/scipharm89020023_

Round 1
Reviewer 1 Report
In this manuscript, Uttarawichien and colleagues investigated the anti-angiogenesis activity of quercetin in colorectal cancer cells and endothelial cells. They demonstrated that quercetin had an anti-angiogenesis effect on VEGF-A inhibition related with the NF-kappaB signalling pathway in the HT-29 cells and reduced VEGFR-2 expression and translocation in the HUVECs.
The manuscript was interesting to read. However, various important points will have to be addressed, as described below:
Major points:
1- Line 66: “Several studies have reported that quercetin has an anti-angiogenic effect in prostate cancer, breast cancer, retinoblastoma, and head and neck carcinoma [22–26]”. The anti-angiogenic effect of quercetin has already been demonstrated in various cancers. Testing its efficacy on colorectal cancer is therefore of limited novelty.
2- The investigation of the anti-angiogenic effect of quercetin in HT-29 cell-induced angiogenesis in both colorectal and endothelial cells through the VEGF-A/VEGFR-2 pathway would probably be more suitable for a pharmacology-related journal.
3- Line 124: “The cells were observed under a fluorescence microscope”: could you please indicate the excitation and emission wavelengths used. Could you please also add a flow cytometry experiment to provide quantitative data supporting the qualitative results obtained.
4- Indirect immunofluorescence assay: qualitative experiment relying on counting the fluorescence in random fields. It would be good to support these results by additional experiments providing quantitative analysis.
5- Line 152: “quercetin dramatically inhibits cells proliferation, whereas low concentrations have no anti-proliferative effect on either cells type”: could you please remove “dramatically” from the manuscript, as the showed results are moderate. Could you please add higher quercetin concentrations to support this claim, as only one high concentration was tested with limited effect.
Minor points:
6- Line 54: “it will translocate into a nucleus”: could you please replace “a” by “the”
Material and methods
7- Line 99: “1.0x104 cells were seeded”: could you please correct the cell number
8- Line 99: “in a 96-well plate”: could you please modify this as “in 96-well plates”
9- Line 100: “atmosphere that was supplied”: could you please reformulate this
10- Line 109: “the cells were seeded”: could you please indicate the cell concentration which was seeded
11- Line 115: “VEGFR-2 expressions”: could you please correct “expression”
12- Line 116: “4x104 cells”: could you please correct the cell number
13- Line 117: “serum free media containing 5 or 10 μg/ml quercetin was added”: could you please correct “were”
14- Line 119: “a primary antibody was added”: could you please name the primary and secondary antibodies, indicate their concentration
15- Line 121: “Hoechst-33342 was used for counterstaining”: could you please provide more experimental details
16- Line 123: “the HUVEC cells were harvested”: could you please provide more experimental details
17-Line 128: “Total protein was obtained from the HT-29 cells treated with quercetin at concentrations of 5 and 10 μg/ml by using a cold RIPA buffer”: could you please describe how the proteins were extracted
18- Line 132: “lysate proteins were load”: could you please correct “loaded” and indicate the amount of proteins loaded
19- Line 133: “the surface was blocked and then probed with primary antibodies and then secondary antibodies”: could you please provide more experimental details
Results
20- Line 168: “quercetin dramatically inhibits”: could you please remove “dramatically” from the whole manuscript
Discussion
21- Line 279: “which could then be uses”: could you please correct “used”
Reviewer 2 Report
The article under review reports on the downregulation of VEGF-A/VEGFR2 in human colon cancer cells (HT-29) and human vein endothelial cells (HUVECs) by the flavonoid quercetin, as a means to evaluate a potential anti-angiogenetic effect of this compound. The title adequately reflects the subject of the research. The abstract is correct and concise, the introduction provides a significant scientific background on the subject. The experimental section describes accurately and sufficiently the materials and methods used in the research. The authors employ standard methods in the investigation of anti-angiogenetic effects on cancer cells. Results are presented clearly and are well illustrated. The discussion chapter makes the analysis of the obtained results. The authors are asked to perform in this section some specific comparisons to the anti-angiogenetic effects obtained on other cell lines (cited in the introduction: prostate cancer, breast cancer, retinoblastoma, head and neck carcinoma) and to comment on the similarities/differences regarding the observed effects at the respective concentrations. The authors could also note that of all the cancer types investigated in this regard, the particular localization of colorectal cancer in the digestive tract enables the achievement of significantly higher doses of quercetin at the cancer site, which is an advantage considering the low bioavailability of this compound. This should underpin the merits of the current research.
